# FGGP: Fixed-Rate Gradient-First Gradual Pruning

Anonymous Full Paper
Submission 36

## Abstract

In recent years, the increasing size of deep learning models and their growing demand for computational resources have drawn significant attention to the practice of pruning neural networks, while aiming to preserve their accuracy. In unstructured gradual pruning, which sparsifies a network by gradually removing individual network parameters until a targeted network sparsity is reached, recent works show that both gradient and weight magnitudes should be considered. In this work, we show that such mechanism, e.g., the order of prioritization and selection criteria, is essential. We introduce a *gradient-first* magnitude-next strategy for choosing the parameters to prune, and show that a *fixed-rate* subselection criterion between these steps works better, in contrast to the annealing approach in the literature. We validate this on CIFAR-10 dataset, with multiple randomized initializations on both VGG-19 and ResNet-50 network backbones, for pruning targets of 90, 95, and 98% sparsity and for both initially dense and 50% sparse networks. Our proposed fixed-rate gradient-first gradual pruning (FGGP) approach outperforms its state-of-the-art alternatives in most of the above experimental settings, even occasionally surpassing the upperbound of corresponding dense network results, and having the highest ranking across the considered experimental settings.

## 1 Introduction

In deep-learning for a given problem setting, typically first a network architecture is engineered (handcrafted) and then the parameters of such network are learned. However, even when the problem setting has an established solution with a known network architecture, the required number of features/filters, contextual depth, layer sizes, and other architectural settings often need to be adjusted empirically to achieve optimal results. Alternatively, overparameterized deep neural networks are used, as most state-of-the-art today, aiming to capture hidden patterns in the data without manually optimizing architectures. Overparameterization, however, comes with largely increased computational costs at both training and inference time; requiring more energy, yielding higher $CO_2$ emissions, and making models less suitable for time critical tasks and embedded/edge/mobile computing. Overparameterized models are also more likely to overfit the training data, hence yielding reduced performance especially without suitable regularization treatments, due to suboptimal optimization approaches or insufficient time required for lengthy training.

Neural Architecture Search (NAS) [1], a type of meta-learning and a subfield of automated machine learning (AutoML), aims to find optimal NN architectures. NAS typically treats networks as black-box models updating them based solely on observed output or prediction accuracy using methods such as evolutionary search, reinforcement learning, Bayesian approximation, etc. This typically requires significant amount of resources and does not seek principled update strategies that consider the intrinsic dynamics and parameters of a network.

Pruning is a network model compression technique that aims to remove the network parameters that are least important, i.e., that would change the accuracy minimally. Ideas of adapting network architectures started as early as the introduction of neural networks themselves, including seminal works such as "Optimal Brain Damage" (OBD) [2] by LeCun et al. For a while, research mostly focused on devising expressive neural representations, on efficient optimization strategies, and on solving practical large problems thanks to advances in compute capability. Recently, pruning has gained popularity again with an increasing focus on model compression for highly complex problems and edge computing.

Our main contributions in this paper include: (1) We provide a clear and transparent definition and review of multi-step top-K selection processes in gradual pruning. (2) We show the order prioritization and selection criteria both being essential and inter-related for a successful gradual pruning algorithm. (3) We propose a gradient-first top-K selection criterion that performs well with a fixed-rate selection quota. (4) We set the new state-of-the-art in gradual pruning for CIFAR-10 dataset.

## 2 Background

Network pruning was shown by Frankle and Carbin [3] and Liu et al. [4] to achieve similar or even better classification performance than corresponding dense models, with less than half of the original parameters. These helped draw further attention to the redundancy of state-of-the-art deep neural networks. *Structured pruning* removes neurons in

fully-connected (FC) or convolutional (conv) layers (the latter also known as *channel pruning*), hence not changing the layer's original structural property, i.e., yielding a respective FC or conv layer. *Unstructured pruning* removes weights (connections), which then typically makes the layer (and the network) *unstructured*, i.e., not a conventional FC or conv anymore.

**Structured pruning** aims to remove redundant channels such as based on LASSO regression [5] and discrimination-aware loss [6], or to remove redundant neurons or convolutional filters such as based on mean activation magnitude [7]. Instead of such (structured) pruning, it is more natural to decide on the (unstructured) pruning of network parameters since these are the entities that are determined via optimization during training.

**Scheduling.** Some methods update the neurons (filters) using a single one-shot approach, either at the very beginning (following initialization) or at the very end (after training to convergence). Pruning at start, also known as *foresight pruning*, assumes that an optimal subnetwork, which is capable of achieving success at convergence, is already identifiable at initialization. For instance, SNIP [8] approximates synapse sensitivity after initialization by estimating the change in loss with respect to the removal of each parameter. To avoid numerous forward passes by removing each parameter individually, SNIP instead makes an infinitesimal (multiplicative) approximation to removal that can be computed in a single forward-backward pass, and then pre-prunes the parameters with small magnitude $|\theta_i|$ and small gradients $|g_i|$ at initial state. Gradient Signal Preservation (GraSP) [9] revisits SNIP by considering expected subsequent gradient flow using a second-order term $|Hg|$, which avoids explicit Hessian computations; however, this leads to results not substantially different than SNIP. Despite the simplicity and attraction of one-shot methods, superior results are often achieved using sequential pruning approaches, indicating that fixing the network structure once is not an optimal strategy.

**Iterative pruning** is applied repeatedly over multiple rounds, following complete convergence after each round, which is hence computationally very costly. Lottery Ticket Hypothesis (LTH) [3] assumes that a pruning-candidate subnetwork has the *winning ticket* primarily thanks to its random initialization of parameters. LTH then trains a network, prunes the weights with smallest magnitudes, and then retrains the pruned network starting from the *same* initial random parameters, and repeats this process iteratively until a desired sparsity level is reached. Later, Liu et al. [4] confute LTH by showing that any arbitrary initialization with such pruned network achieve similar results, hence showing that the key is the architecture, not the initialization.

Fixed-sparsity pruning initializes a network at the target low sparsity and then trains this network to convergence while keeping its sparsity constant, such that the total training cost can be kept lower than a dense network. RigL [10] is such an example, which during training first selects a subset of weights with the smallest magnitudes to prune, and then momentarily sets all missing weights to zero to compute their gradient with a backprop, to determine the highest gradient-magnitude weights to add (grow) to keep the sparsity constant.

**Gradual pruning** prunes the network while it is being trained, slowly changing the network sparsity to a final targeted value, e.g. at regular iteration or epoch intervals some parameters are pruned based on a priority criterion and mechanism. This was shown to achieve comparable performance to iterative pruning, while incurring much lower computational costs. The main challenge here is that the network is not in a converged state during the pruning decisions. Medeiros et al. [11] prune connections with lower correlations between the errors within a layer and those backpropagated to the preceding layer, which they call the MAXCORE principle. Dynamic Network Surgery [12] employs gradual pruning with a binary mask for pruned/spliced connections, while updating both the pruned and the remaining parameters. Zhu et al. [13] gradually change the network sparsity based on a cubic scheduling function, while pruning the weights with smallest magnitudes – although the weights alone are not sufficiently informative in an uncoverged network state. Dettmers et al. [14] utilize exponentially smoothed gradients (momentum) to identify layer and parameter contributions to error reduction, while both pruning and regrowing the connections based on momenta. GraNet [15] combines the pruning schedule of [13] with the pruning criterion of RigL [10], achieving the state-of-the-art results in unstructured gradual pruning. Note that although GraNet calls the subset selection process as weight "addition" (where the second stage is explained as if adding [back] high-gradient parameters), this is somewhat a misnomer as GraNet does not aim and cannot grow synapses inexistent at the beginning of pruning. In this paper, we describe GraNet with a literature-consistent terminology, which helps to better contrast it with our proposed method.

# 3   Methods

For a neural network $f$ parametrized by $\Theta = \{\theta_1, \theta_2, ..., \theta_w\}$ with $w$ parameters, the goal of training on a dataset $\mathcal{D} = \{(x_1, y_1), (x_2, y_2), ..., (x_K, y_K)\}$ with $K$ input-groundtruth pairs $(x_i, y_i)$ is

$$\min_{\Theta} L = \sum_{i=1}^{K} l(f(x_i; \Theta), y_i), \qquad (1)$$

where $l(\cdot, \cdot)$ is the penalty/loss function for the distance between $y_i$ and the network prediction $f(x_i; \Theta)$. The optimization problem is then solved iteratively via backpropagation (training), which is typically stabilized by using a regularization of parameters as an additional objective.

## 3.1 Pruning schedule

Let sparsity $s$ define the number of parameters, $N$, in a pruned network with respect to its dense equivalent $N^*$ as $N = (1 - s)N^*$. Gradual pruning reduces the network parameters slowly over training time based on a desired decay pattern (also called "schedule"). To prune a network from an initial sparsity $s_{\mathrm{ini}}$ at iteration $t_{\mathrm{ini}}$ to an intended target sparsity $s_{\mathrm{fin}}$ at iteration $t_{\mathrm{fin}}$, we employ cubic sparsity scheduling [13] where sparsity $s_t$ at iteration $t$ is given by:

$$s_t = s_{\mathrm{fin}} + (s_{\mathrm{ini}} - s_{\mathrm{fin}})(1 - \frac{t - t_{\mathrm{ini}}}{t_{\mathrm{fin}} - t_{\mathrm{ini}}})^3, \quad (2)$$

which then defines the desired number of parameters at any iteration $t$ as $N_t = (1 - s_t)N^*$.

Pruning events can either be applied regularly during training, e.g., every $\Delta t$ epochs or iterations, or be at instances sampled randomly from a probability distribution. Each event will then prune $N_p$ network parameters to reduce their number to that desired (scheduled) at that instance, i.e., $N_p = N_{t-\Delta t} - N_t$ assuming pruning events with $\Delta t$ iteration interval.

## 3.2 Pruning strategy

Parameter selection criteria and mechanism have the utmost importance that can affect the outcome significantly. We motivate our choice based on the framework of OBD [2], which approximates the sensitivity of loss $L$ to individual network parameters $\theta_i$ using a second-order Taylor-series expansion as:

$$\delta L = \underbrace{\sum_i g_i \delta\theta_i}_{\text{1st term}} + \underbrace{\frac{1}{2}\sum_i h_{ii}\delta\theta_i^2}_{\text{2nd term}} + \underbrace{\frac{1}{2}\sum_{i \neq j} h_{ij}\delta\theta_i\delta\theta_j}_{\text{3rd term}} \quad (3)$$

where higher order terms are omitted, $g_i$ is the gradient of $L$ with respect to $\theta_i$, and $h_{ij}$ are the elements of the Hessian matrix. Pruning a parameter, which nullifies its effect, causes a negative change equivalent to its value, i.e., $\delta\theta_i = -\theta_i$. The 3rd term is often omitted by assuming minimal cross-parameter effect. If the network is already trained (i.e., converged at a local minimum), then the gradients diminish and the 1st term can be omitted as well. OBD then estimates the diagonal Hessian terms to prune parameters based on the 2nd term.

The above, however, cannot be assumed in a gradual pruning setting where the network is not converged. Assuming a simpler first-order Taylor expansion $\delta L \approx \sum_i g_i \delta\theta_i$, several works aim to minimize this by simply pruning parameters with small magnitudes, but this only applies if those gradients are not large. Although some recent works [8, 10, 15] consider the gradients in addition, they do this without a basis on the terms higher than the first order. In this work, we consider the gradients first, focusing on the parameters with small gradient magnitudes for which the 1st term in (3) has a basis to be omitted, and then we focus on small magnitudes that ensure the 1st term to diminish as well as the 2nd term where they appear quadratically – selecting the parameters with minimal effect on the loss.

A pseudocode of our proposed approach fixed-rate gradient-first gradual pruning (FGGP) is given in Algorithm 1, with the pruning criteria visualized in Figure 1(a). At every $\Delta t$ iterations, our method chooses the parameters to prune with a two-step selection process: We first rank the parameters by their gradient magnitudes $|g_i|$; we select the smallest $rN_{t-\Delta T}$ out of these, and then rank those by their parameter magnitude $|\theta_i|$; finally we select the smallest $N_p$ of these to prune. This strategy avoids the magnitude-based selection from applying to unconverged parameters, whose values are still being changed, i.e., having large gradient magnitudes. GraNet, in contrast, applies an opposite order of

---

**Algorithm 1** FGGP algorithmic overview

**Inputs:**
1: network $f_\Theta$, dataset $D$
   **Initialize**
2: Neural network $f_\Theta$
3: $s_t$ : sparsity scheduled as in (2)
4: $\Delta t$ : update interval
5: $r$ : sub-selection rate
6: **for** each training iteration $t$ **do**
7:     Sample a minibatch $B_t \sim D$
8:     **if** $t \equiv 0 \pmod{\Delta t}$ **then**
9:         Sort the $N_{t-\Delta t}$ parameters in ascending order by gradient magnitude (step 1 in Figure 1(a))
10:         For the first $r \cdot N_{t-\Delta t}$ parameters, sort in ascending order by weight magnitude (step 2)
11:         Prune the first $N_{t-\Delta t} - N_t$ parameters, so there are $N_t$ parameters left (step 3 in Figure 1(a))
12:     **end if**
13:     Update parameters via backpropagation
14: **end for**

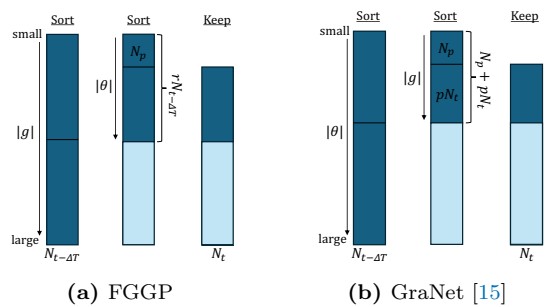

**(a)** FGGP      **(b)** GraNet [15]

**Figure 1.** Comparison of the parameter selection mechanisms between our proposed FGGP and GraNet [15].

selection as illustrated in Figure 1(b), which may fail by pruning important parameters if many parameters with small magnitudes are still in change (i.e., with large gradients), since its 2nd step can only consider those culled/selected from the 1st step. With our subset selection order, we avoid this.

For pruning we consider the paremeters from all the network layers, unlike RigL [10] which does not prune the first convolutional layer and LTH [3] which does not prune the last fully-connected layer. We prune the network *globally*, pooling the parameters from all layers for their prioritization in pruning.

## 3.3 Subset selection rate

In the above process, an important factor is the decision of which gradients to consider as large to omit from the next steps of parameter selection. Any fixed threshold would not be applicable, since the parameters and gradients all have values relative to each other, and in gradual pruning a predetermined schedule has to be met to achieve a desired sparsity. So, the selection needs to be based on a ratio from a ranked (sorted) prioritization. In its magnitude-first strategy, GraNet employs (cosine) annealing to reduce a parameter $p$, seen in Figure 1(b). This reduces the subset considered from step 1 over training iterations, focusing on incresingly smaller parameter magnitudes at later iterations. In our gradient-first strategy, such reduction is not necessary and is found to be counterproductive in our ablation studies. Instead we utilize a fixed rate $r$ as the ratio of gradient magnitudes to selected from the first step. We herein set $r = 0.5$ such that the parameters with gradient magnitudes smaller than their median value are taken into further consideration.

## 4 Experiments and Results

For evaluation we use the CIFAR-10 dataset as common in the field, allowing us to compare our results to multiple published works. We evaluate our method based on ResNet-50 and VGG-19 architectures, as were adapted for the CIFAR dataset [4]. Implementation details are given in Appendix A. We aim for target sparsities $s_{fin} = \{90, 95, 98\}$ in two experimental settings of dense-to-sparse ($s_{ini} = 0\%$) and sparse-to-sparse ($s_{ini} = 50\%$). For initializing the parameters in sparse networks, we employ Erdős–Rényi Kernel (ERK) [10] inline with the compared state-of-the-art. See Appendix B for details.

In comparisons, we provide single-shot pruning results from other methods as reference. The pruning accuracies of the methods with a dense network upperbound are seen in Table 1, where $\pm$ results indicate those from three different random initialization. The single-shot methods are seen to be inferior to the gradual methods, and among the latter the ones

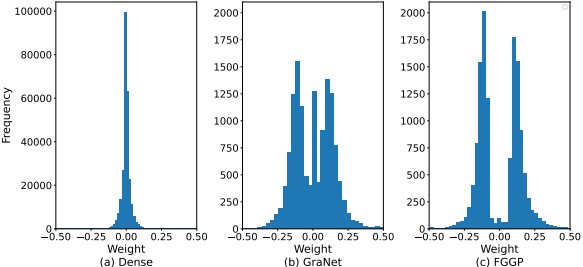

**Figure 2.** Comparison of weight magnitude distributions at the end of training with (a) a dense network, as well as pruned with (b) GraNet and (c) FGGP from 0% to 95% sparsity.

that consider both gradient and parameter magnitudes (e.g., RigL, GraNet, and our FGGP) perform relatively better. These differences are more pronounced at higher sparsity targets and for ResNet-50 architecture, indicating that these potentially represent more challenging pruning scenarios. In most scenarios our method FGGP outperforms the other state-of-the-art methods. For VGG-19, our method is more successful at higher sparsity targets of 95% and 98%, for both sparse- and dense-to-sparse training scenarios. In both scenarios for ResNet-50, two-out-of-three sparsity levels our method outperforms the others – although some results may be too close to call for a clear winner. For an overall comparison across all experiments, we employ a ranking strategy where the methods are ranked by their mean accuracy in each experimental configuration (each column and grouping). Our method is seen to lead the overall rankings.

For a sparsified network, the distribution of number of parameters across layers exemplified for VGG-19 in Appendix C indicates that the depth of this network was potentially redundant for the given task, as most filters in the latter half of the layers have been sparsified almost completely, without much reducing the accuracy as seen in our results. To provide further insight, in Figure 2 we present the distributions of weight magnitudes for networks trained using a dense model as well as using GraNet and the proposed FGGP. Although both pruning methods reduce near-zero weights, which may have less impact on the final predictions, our approach is seen to be more effective in this regard.

### 4.1 Ablation Study

To study the effect of the proposed method components and parameters, we conduct ablation experiments for the dense-to-sparse setting with VGG-19 on CIFAR-10, repeating each experiment with three initialization seeds and reporting the mean values for comparison.

First, we evaluate the impact of the subset selection rate $r$ by comparing results for values $r =$

**Table 1.** Test accuracy of pruned VGG-19 and ResNet-50 networks on CIFAR-10 dataset, with mean±std values from experiments with three different seeds. GraNet [15] and RigL [10] results were taken from [15], while the other results were compiled from [9, 16] for the reported settings. The bold numbers indicate the best accuracy in each given gradual pruning subcategory. The rightmost column shows the average ranking of methods within a subcategory across the six experimental settings (columns) (the stars indicate averages from VGG-19 only).

| | | VGG-19 | | | ResNet-50 | | | Rank |
|---|---|---|---|---|---|---|---|---|
| Sparsity target (N%) | | 90% | 95% | 98% | 90% | 95% | 98% | |
| Single-shot (0%→N% at init) | SNIP [8] | 93.63 | 93.43 | 92.05 | 92.65 | 90.86 | 87.21 | 2.17 |
| | GraSP [9] | 93.30 | 93.04 | 92.19 | 92.47 | 91.32 | 88.77 | 2.33 |
| | SynFlow [17] | 93.35 | 93.45 | 92.24 | 92.49 | 91.22 | 88.82 | 1.50 |
| Sparse-to-sparse (50%→N% gradual) | Deep-R [18] | 90.81 | 89.59 | 86.77 | - | - | - | 5.00* |
| | SET [19] | 92.46 | 91.73 | 89.18 | - | - | - | 4.00* |
| | RigL [10] | 93.38±0.11 | 93.06±0.09 | 91.98±0.09 | 94.45±0.43 | 93.86±0.25 | 93.26±0.22 | 3.00 |
| | GraNet [15] | **93.73±0.08** | 93.66±0.07 | 93.38±0.15 | 94.64±0.27 | **94.38±0.28** | 94.01±0.23 | 1.67 |
| | FGGP (ours) | 93.68±0.04 | **93.94±0.17** | **93.63±0.15** | **94.76±0.11** | 94.27±0.38 | **94.22±0.24** | **1.33** |
| Dense-to-sparse (0%→N% gradual) | STR [20] | 93.73 | 93.27 | 92.21 | 92.59 | 91.35 | 88.75 | 4.50 |
| | SIS [16] | **93.99** | 93.31 | 92.16 | 92.81 | 91.69 | 90.11 | 3.67 |
| | GMP [21] | 93.59±0.10 | 93.58 ±0.07 | 93.52±0.03 | 94.34±0.09 | 94.52±0.08 | 94.19±0.04 | 3.17 |
| | GraNet [15] | 93.80±0.10 | 93.72±0.11 | 93.63±0.08 | 94.49±0.08 | 94.44±0.01 | **94.34±0.17** | 2.00 |
| | FGGP (ours) | 93.71±0.15 | **93.77±0.25** | **93.80±0.02** | **94.78±0.19** | **94.64±0.38** | 94.33±0.42 | **1.67** |
| Dense (0% upperbound) | | 93.93±0.35 | | | 94.73±0.06 | | | |

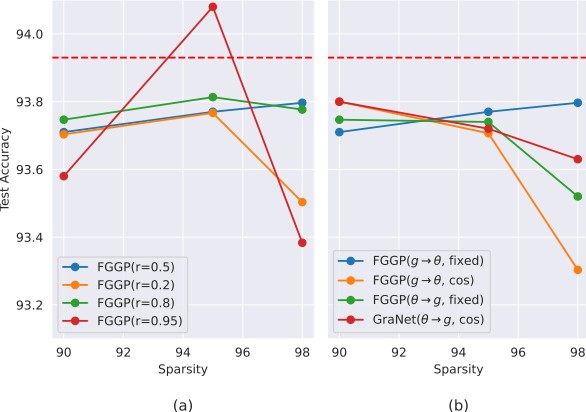

**Figure 3.** (a) Comparison of FGGP for four different subset selection ratios $r$. (b) Ablations of FGGP indicated as $(\cdot,\cdot)$ where the first indicated the pruning criteria order ($g{\to}\theta$: gradient-first & $\theta{\to}g$: magnitude-first) and the second the subset selection strategy (with the rate *fixed* or varying as *cos*ine annealed). GraNet's proposed method choices are also indicated in the same notation for clarity. Note that FGGP($g{\to}\theta$, fixed) is our proposed mechanism with gradient-first fixed-rate subset selection. Experiments are reported for dense-to-sparse pruning of VGG-19 for target sparsities of {90,95,98}%.

$\{0.20, 0.50, 0.80, 0.95\}$. Note that in the extreme case of $r = 1$, the second stage would be the sole determining criterion -— effectively reducing the method similar to gradual magnitude pruning (GMP). The results are depicted in Figure 3 (a). Although settings differ in their performance (with $r = 0.95$ even surpassing the upper bound at 95% sparsity), overall $r = 0.5$ and $r = 0.8$ consistently perform well. We chose $r = 0.5$ for all the experiments given its superior trend with higher sparsity, as a primary target of network compression.

Second, we ablate different parts of our method FGGP, mimicking the GraNet behaviour to assess the components with positive contribution. For the subset selection, we use our fixed rate as well as the varying (cosine annealing) rate change from GraNet. We also test the change of order from gradient-first to magnitude-first, as well as the combinations of this with the rate choice above. The results are seen in Figure 3(b). As the sparsity level gets higher, our proposed method FGGP with the gradient-first and fixed-rate settings are seen to achieve the best performance. Note that for all these methods the cubic scheduling function already reduces $N_p$ over time, so the results may indicate that an additional reduction of the subset selection rate is redundant and detrimental.

# 5    Discussion and Conclusion

In this paper, we consider both gradient and weight magnitudes in the unstructured gradual pruning of parameters to sparsify networks. We herein argue that the criteria and the mechanism (the order, thresholds, etc) used in the prioritization of pruned parameters are essential. We propose to use a fixed-ratio of parameter gradient magnitudes as a first decision criteria for pruning, and experimentally validate this in a variety of settings. Pruning has the potential to substantially reduce computational costs in deep learning, thereby contributing to lower energy consumption and carbon emissions without sacrificing ultimate performance. Lower energy use can enable novel end-user experience, such as rendering IoT devices feasible. Having smaller models with similar performances could allow the deployment of complex models on smaller hardware and of very large/deep models that would otherwise be infeasible.

Note that Liu et al. [15] explain their method GraNet as being able to regenerate/add ($pN_t$) connections/parameters to a network during the pruning operations. However, following their descriptions and pseudocode it becomes evident that they apply a two-step strategy which first ranks the parameters by their magnitude $|\theta|$ to select the subset of smallest $N_p + pN_t$ for further ranking gradient magnitude $|g_i|$ to select the smallest $N_p$ of them to prune, as demonstrated in Figure 1(b). We believe this demystification of such state-of-the-art is a further minor contribution of our work, as it also enabled us herein to technically compare our methods and experimentally design comparative ablation experiments. Note that at any given training step, some parameters may momentarily be zero (e.g., while changing sign) despite having nonzero gradients. If the number of such parameters exceeds $N_p + pN_t$, the second step of GraNet may inadvertently prune essential parameters, especially for larger $p$ at earlier iterations. In contrast, our approach prioritizes gradient magnitudes in the first ranking step, ensuring that only relatively stable parameters (those that have locally converged) are considered for pruning. Note that our above criterion is more conservative than GraNet, and it may disregard some good parameter candidates. Also, if all gradients are large, changing parameters may still be considered erroneously. Nevertheless, the results show that our strategy is superior to that of the earlier state-of-the-art. In the future, including $|g\theta|$ and/or Hessian approximations can potentially improve the results further.

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

# A  Experiment details

Our proposed method FGGP is implemented in Pytorch 2.0. We use cross-entropy loss for classification and the Stochastic Gradient Descent (SGD) for optimization in all experiments. Training hyperparameters are tabulated in Table A.1. We set $\Delta t = 1000$ iterations. Pruning is stopped after 80% of the targeted epochs (i.e., $t_{fin}$ is set to the iteration number forecast for the 148th epoch), hence leaving the remaining 20% of training to fine-tune the model parameters without any disruption from architectural changes – a strategy also common in other gradual pruning approaches. CIFAR-10 dataset contains 10 different classes representing airplanes, cars, birds, cats, deer, dogs, frogs, horses, ships, and trucks. It consists of 60k 32×32 color images with 6k images for each class, with a split of 50k training and 10k testing images. We use data augmentation with random crops with padding of 4 and horizontal flips.

# B  Erdős–Rényi initialization

Erdős–Rényi [19] is a strategy for initializing the parameters in fully-connected layers. Erdős–Rényi Kernel (ERK) [10] offers an extension of this to convolutional layers. Such initialization was shown to perform superior to networks initialized randomly [10]. ERK [10] determines a factor $f_l$ to scale the initialization of the parameters in the convolutional kernel $l$ with width $w_l$ and height $h_l$ as:

$$f_l = 1 - \frac{n_{l-1} + n_l + w_l + h_l}{n_{l-1} \times n_l \times w_l \times h_l}, \quad (4)$$

where $n_l$ is the total number of parameters in that convolutional kernel.

# C  Sparsity of pruned network

To give an insight of where the parameters are pruned the most and how the pruned networks look like, in this section we show the sparsity distribution of networks after applying FGGP. The number of parameters of a dense and an FGGP pruned VGG-19

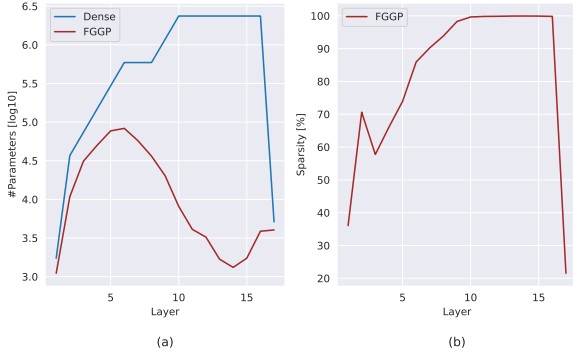

**Figure C.1.** (a) Number of parameters per layer in a dense and FGGP-pruned VGG-19 network, shown in logarithmic scale. (b) Sparsity of each layer after pruning. The results are shown for a sample experiment. Note that the final layer is fully-connected while the others are convolutional.

are shown in Figure C.1(a), with the resulting layer sparsity plotted in Figure C.1(b). As can be seen, the second half of the network is almost completely sparsified, likely leaving one or a few unit filters to simply forward propagate the information extracted by the initial convolutional layers for the final the prediction in the last fully-connected layer (which hence could not sparsify much). This observation suggests that the depth of VGG-19 may be highly redundant for the task of CIFAR-10 classification, which could potentially be tackled with a half-the-depth network. Future studies shall investigate this aspect, potentially using pruning as an automatic tool to determine optimal network shape and size of traditional handcrafted architectures.

**Table A.1.** Training hyperparameters.

| Data | CIFAR-10 |
|---|---|
| Model | VGG-19 / ResNet-50 |
| Epochs | 160 |
| Batch Size | 128 |
| LR | 0.1 |
| LR Decay Epoch | [80, 120] |
| LR Decay Factor | 0.1 |
| Weight Decay (L2) | 0.0005 |
| Momentum | 0.9 |

