# OpenReview forum: "FGGP: Fixed-Rate Gradient-First Gradual Pruning"
_NLDL.org/2025/Conference — Submitted to NLDL 2025_

### Official Review · Reviewer_TQmq · 2024-10-07
**Review for "FGGP: Fixed-Rate Gradient-First Gradual Pruning"**

**Confidence:** 4

**Summary:**

The paper presents a method for performing neural network pruning. It relies in both considering the gradient value and the absolute weight value for pruning in a two-step fashion. The proposed approach is then compared to a few approaches from the literature.

**Strengths:**

-The Background section is especially thorough.

-Both the presentation of the algorithm and the experimental section are clear and sufficiently detailed. The experiments include a variety of pruning thresholds and benchmarks.

**Weaknesses:**

1. I feel like the theoretical contribution is somewhat thin. When compared to GraNet, the proposed approach keeps the two-step fashion where one considers the absolute value of the weights, while in the other step, the gradient of the weights is considered. The sorting is applied for similar purposes. Thus even though Section 3.2 explains thoroughly the choice in the various steps of the algorithm, I feel like most of the theoretical contribution has already been done somewhere else.

2.1. The experimental section does not lead to the conclusion that the proposed approach has a significant impact on the obtained performances. It is said that « We herein argue that the criteria and the mechanism (the order, thresholds, etc) used in the prioritization of pruned parameters are essential » (line 390), yet Table 1 does not depict this specific choice of order in prioritization as « essential ».

2.2. It can be found that in most of the experiments, FGGP is the method leading to the biggest variance, sometimes a few times bigger than the runner-up. Considering this, and the fact that only 3 random seeds were used for the results presented in Table 1, I think using many more random seeds is necessary.

**Typos and such**

-Lines 78 and 96 : please use inline citation syntax, as in Line 92.

-Lines 92 and 110 VS 136 and 151 : please be consistent in how subsection titles are used.

-It would be best to name the terms from Equation (3), for « the first term » or « the second term » are confusing names.

-Line 322 : initiatlization

-Though what FGGP stands for is in the title of the work, please present the name and the abbreviation in due form.

-Line 376 : text → test

**Justification:**

The theoretical contribution is somewhat thin, and the experimental section (though the diversity of experiments is remarkable) is insufficient to conclude in any significant empirical gains from the proposed approach (see **Weaknesses**).

---

> ### Author Rebuttal · Authors · 2024-10-25
>
> We thank the reviewer for the feedback. We have incorporated all the typographic suggestions.
>
> > “Table 1 does not depict this specific choice of order in prioritization as « essential »”
>
> In Table 1, we present the performance of different pruning methods across various network structures and sparsity levels; therefore, it does not include information about prioritization. To emphasize the importance of prioritization (e.g., gradient- or magnitude-first), we have now included a histogram of weights in Fig. 2 of our revised manuscript. This figure illustrates that different prioritization strategies together with selection ratios during pruning may lead to distinct final weight distributions, such as being more closely centered around zero or farther from zero.
>
> >  “It can be found that in most of the experiments, FGGP is the method leading to the biggest variance, sometimes a few times bigger than the runner-up.”
>
> While it is true that the variance with our proposed method is sometimes higher than that of runner-up state-of-the-art method (4 out of 9 experimental setups), in the other settings the opposite is true.
>
> >  “Considering this, and the fact that only 3 random seeds were used for the results presented in Table 1, I think using many more random seeds is necessary.”
>
> Most literature on pruning typically applies only three different seeds or even just one; thus, we opted for three seeds in our study.  Also, we have chosen three seeds, as this was the number also in the compared methods and more seeds would increase the computation time, especially since several steps of gradual pruning iterations are needed for each.

---

### Official Review · Reviewer_ZXiL · 2024-10-07
**This paper proposes a new pruning strategy FGGP, but with very limited empirical evidence or theoretical proof.**

**Confidence:** 5

**Summary:**

While magnitude pruning is a very established method, the papers argues that if the gradients of weights are large, those weights are still contributing to the loss, and magnitude pruning alone may not be effective at minimizing the loss. Based on this rationale, this paper proposes a two-step pruning algorithm that considers gradient magnitude before magnitude. Specifically, it first creates a subset of weights by selecting those with the smallest gradient magnitudes and then applies magnitude pruning within this subset. As a result, it omits weights with large gradient magnitude for pruning, even if they have the smallest magnitudes (which traditional magnitude pruning would prune). The method is evaluated by measuring the accuracy on CIFAR-10 using VGG-19 and ResNet-50. The results show situational marginal improvement over several other gradual pruning and sparse training methods.

**Strengths:**

- The paper introduces a unique two-step pruning method that prioritizes gradient before magnitude. While the justification may be limited, the idea itself is a creative attempt to determine the saliency of weights.
- The description of the proposed method is very clear and easy to follow.

**Weaknesses:**

Introduction reads very 'scattered', without emphasis on what the problem is, where the gap is, and what problem the paper is proposing to resolve. It lacks a sumary of key contributions, typically listed at the end of introduction for such paper.

Core (Method+Conclusion):
- "Assuming a simpler first-order... several works aim to minimize this by simply pruning parameters with large magnitudes, but this only applies if those gradients are not large" - they actually remove the smallest magnitude weights (not largest), as they contribute the least to output. Also, stating magnitude pruning only works when gradients are small is oversimplified because it's based on the assumption that small magnitude weights have small contribution to the network, regardless of gradient.
- The author seems to have a misunderstanding of GraNet - in method (Fig 1b) and conclusion, the paper seems to claim GraNet uses a two-step ranking strategy, similar to what this paper proposes, but in reverse - magnitude first, gradient second. However, to the best of my understanding, it prunes by magnitude and regrows by gradient - these are two separate processes. Please refer to the original GraNet paper "We prune the weights with the smallest magnitude, as it has evolved as the standard method when pruning happens during training" and "Again, we use the gradient as the importance score for regeneration, same as the regrow method as used in RigL"
- While the core idea that gradient magnitude first ensures weights contributing more to the loss are protected from pruning is somewhat reasonable in theory, the paper fails to provide a thorough argument for why sorting by gradient first is better than the other way around. Why is this sorting algorithm even neccesary?

Results:
- Results are only marginally better sometimes, and it is not clear whether it is just by chance. Overall, the results are unconvincing.
- A 2024/2025 pruning paper should include at least some form of imagenet, tinyimagenet benchmarks. It needs more than CIFAR-10.
- Ablation study a is well designed - the author picks r={0.20, 0.50, 0.80, 0.95} for the gradient-based subset cutoff, which is then used as a pool for magnitude pruning. However, Figure 2a fails to convey a story. For instance, at 95% sparsity, it's red>green>blue>orange, but at 98% sparsity it's blue>green>orange>red. Red(r=0.95) approachs an extreme case which resembles gradual magnitude pruning, performs much better at 95%. This seems contradictory to your hypothesis. The author then claims r=0.5 performs the best overall - but is it? It is worse than r=0.8 at both 90% and 95%, and only marginally better at 98%. If r=0.95, or 1 (this should be tested as well) is better, doesn't it just kill the novelty of this paper?
- Ablation b once again fails to provide a good narrative. The results are different at different sparsities - it lacks consistency. Even at the same sparsity, it is hard to conclude anything. For example, at 98%, it goes gradient-first, fixed > magnitude-first, fixed > gradient-first, cos. We see that gradient-first, cos is the worse than magnitude-first, so why gradient-first? Then again 95% has a completely different order. The author puts emphesis on the 'order of prioritization', but fails to provide any support here.

Background:
- Background is too long. It has a whole section on structured pruning when the proposed method is exclusively unstructured. Since the author is not re-inventing scheduling, the literature on this my be excessive.
- Background contains technical errors. For instance, " RigL [16] uses a two-step pruning criterion by first selecting a subset of weights with the smallest magnitudes and second selecting a subset therein with the smallest gradients to prune. " - it is not a 2-step criterion, but simply simultaneously prunes by magnitude and regrows by gradient at pre-defined intervals. Moreover, this is not a pruning paper, but sparse training where the sparsity stays uniform throughout training. " Another example, " GraNet [17] combines the pruning schedule of [14] with the pruning criterion of [16], achieving the state-of-the-art results in unstructured gradual pruning. Note that although GraNet calls the subset selection process as weight “addition” (where the second stage is explained as if adding [back] high-gradient parameters), this is somewhat a misnomer as GraNet does not aim and cannot grow synapses inexistent at the beginning of pruning" - GraNet indeed adds weights back, refer to Figure 1 of the paper. Although GraNet is used as a key benchmark method in this paper, the author does not seem to grasp key concepts of this method or dynamic sparse training in general.

**Final Rebuttal Confidence:**

5

**Final Rebuttal Justification:**

The author has addressed some of my earlier points, so I have adjusted the original score. However, it is highly advisable to conduct additional experiments to further support this empirical study.

**Justification:**

The decision to prune within a subset of weights with smallest gradients aims to reduce the risk of pruning important weights. However, the effectiveness of this strategy is not well supported by strong theoretical or empirical evidence in the text. It needs much stronger comprehensive evidence such as running on CIFAR-100, TinyImagenet or ImageNet. Moreover, to the best of my understanding, the paper also has technical inaccuracies and misunderstandings in the interpretation of other papers.

---

> ### Author Rebuttal · Authors · 2024-10-25
>
> We thank the reviewer for their appreciation of the unique and creative aspects of our work and the detailed review.
>
> > “The author seems to have a misunderstanding of GraNet - in method (Fig 1b) and conclusion, the paper seems to claim GraNet uses a two-step ranking strategy, similar to what this paper proposes, but in reverse - magnitude first, gradient second. However, to the best of my understanding, it prunes by magnitude and regrows by gradient - these are two separate processes.”
> > “GraNet indeed adds weights back, refer to Figure 1 of the paper. Although GraNet is used as a key benchmark method in this paper, the author does not seem to grasp key concepts of this method or dynamic sparse training in general.”
>
> Firstly, we kindly disagree with all the statements implying that we have misunderstood the literature, especially GraNet, and those stating that it has a separate regrowing process that adds weights to the network.
> To make our point, we start by establishing a clear definition of weight **addition** or regrowth: This term refers to the creation of a new connection (synapse) between a pair of neurons in the network (i.e., introducing a new parameter to be optimized in the model). For such a previously-inexisting parameter, the initial value would naturally not be known or exist. Obviously, one can then not make a pre-addition decision on such an inexistent parameter, nor its gradient, etc, as the GraNet method implies.  Indeed, although one may be confused and unintentionally misled by the text and Fig. 1 in the GraNet paper, it becomes clear from their Method description around Equations 2 and 3, as well as in Algorithm 1 in their supplementary material, that GraNet involves no actual addition but rather a two-step pruning process.
> To be more specific: In the GraNet paper, Eq 2 explains the removals and Eq3 are the so-called "additions". The latter starts with saying "Immediately after that, we regenerate", i.e., no other operation (training, parameter update, etc) is performed in-between the two steps. Thus, the “regenerated” parameters are the very same ones that were so-called removed, hence in effect, they are simply untouched. We also checked this and confirmed in the code that they provide online. Furthermore, Eq3 is using the gradient magnitudes for selection, which would not exist if these edges/variables had truly been removed.
> With our (and the common) definition, one would only call an operation an addition, if the values of the parameters are forgotten.  Otherwise, with a further abuse of terminology, one could state that “We are removing all network weights and then adding and regrowing them all back” by simply doing nothing.
>
> Therefore, regardless of how it was called ("addition" or "neuroregeneration" or else) the above operation is not an addition per common understanding, since nothing has truly been removed at that point. Instead, it is a filtering of what should not be removed by its subsequent step. We realize that the choice of terminology in their paper, despite its promise, can be confusing and misleading. That is why, indeed, in our paper we do our best to elaborate on how such ranking-based filtering processes can work in general, aiming for a precise and transparent terminology. We are therefore confident about our understanding of the field and the GraNet paper.
>
> > “Results are only marginally better sometimes, and it is not clear whether it is just by chance. Overall, the results are unconvincing.”
>
> Similar marginal improvements have indeed justified several other successful publications in the computer vision and deep learning literature. For a close example, GraNet results in their Table 2 also show rather marginal improvements over their closest competitors. Note that our results are averages over multiple initializations, which provides a more robust comparison to several single-init methods in our comparison table, also demonstrating our intention to minimize the element of chance.
>
> > “It lacks a sumary of key contributions, typically listed at the end of introduction for such paper.”
>
> In order to emphasize our contributions, we have now included a ‘summary of key contributions’ as suggested.
>
> > “Also, stating magnitude pruning only works when gradients are small is oversimplified because it's based on the assumption that small magnitude weights have small contribution to the network, regardless of gradient”
>
> We are not sure if we understand this critique, but our related statement is based on the following reasoning: most magnitude pruning methods prune weights at a converged state, where the gradient magnitudes are near zero. As a result, they can safely assume that small-magnitude weights have a minimal contribution to the loss. However, in (dynamic) gradual pruning, the network is not yet converged during training. Therefore, even though a weight’s magnitude may be momentarily small, if it has a large gradient, it shows a trend that its contribution to the loss would likely become larger in future training steps. This supports our claim that magnitude pruning is most applicable in cases with small gradients, e.g., when a model is trained to convergence.  That’s why for pruning prior to convergence, other heuristics are used as been proposed.
>
> > “The author then claims r=0.5 performs the best overall - but is it? It is worse than r=0.8 at both 90% and 95%, and only marginally better at 98%. If r=0.95, or 1 (this should be tested as well) is better, doesn't it just kill the novelty of this paper?”
>
> While r=0.95 performs well at a sparsity level of 95%, its subpar performance at other sparsity levels limits its overall utility and highlights the potential performance limitations associated with magnitude-based pruning (especially as r approaches to 1). Indeed, setting r=1 would be equivalent to pruning based on magnitude alone, similarly to Global Magnitude Pruning (GMP), which our method is already seen to outperform in Table 1. Note that using r=1 would have also invalidated the use of gradient magnitudes in GraNet, which is clearly not the case.  Our reasoning for using r=0.5 is based on its performance at the highest sparsity ratio, with its trend not showing a reduction in the last sparsity increase either (since network compression is mainly the target of pruning), while it also maintains good performance across other tested sparsity levels. We have now better phrased our reasoning behind this parameter choice.
>
> > “...the paper fails to provide a thorough argument for why sorting by gradient first is better than the other way around. Why is this sorting algorithm even neccesary?”
>
> It would be indeed nice to present a mathematical theory on this, which we do not.  Nevertheless, the same argument above could also be made for the heuristic choice of pruning strategy by GraNet and others. Similarly, we thus resorted to empirical validation of our proposal.
> To provide further insight in the network outputs, we have now added the distributions of weight magnitudes at the end of training, as Fig. 2 in our revised manuscript. Although both pruning methods reduce near-zero weights, which may have less impact on the final predictions, our approach is seen to be more effective in this regard. Although it is not clear how very small non-zero weights may still affect the results, or how the chosen process may interact with the backpropagation algorithms, bath size, etc, the presented results are not only to highlight our approach, but also to show the need for further research and development in this field.
>
> > “[RigL] is not a 2-step criterion, but simply simultaneously prunes by magnitude and regrows by gradient at pre-defined intervals. Moreover, this is not a pruning paper, but sparse training where the sparsity stays uniform throughout training.”
>
> In contrast to GraNet, RigL truly adds (inexistent) weights to the network (by computing grads with a backprop to determine which ones to keep, then setting them to zero once added to the network). We have rephrased and clarified its description in the introduction.
> Although the reviewer wrote “not a pruning paper” in the second sentence, the previous sentence reads “prunes by magnitude,” which indicates to us an implicit agreement that RigL is a pruning paper.  It is true that by keeping the sparsity fixed, it is not a network compression paper, but it still involves pruning. With an analogy to a tree (where the term pruning originates from), if you are cutting some branches, while the others are growing, one still calls this “pruning the tree.”
> To clarify this, we have now expanded this fixed-sparsity category and modified the text in the introduction accordingly.
>
> >“Background is too long. It has a whole section on structured pruning when the proposed method is exclusively unstructured. Since the author is not re-inventing scheduling, the literature on this my be excessive.”
>
> We believe the literature on scheduling is crucial, since this is a key component of any gradual pruning approach. We have, nevertheless, shortened the text on structured pruning (which is indeed beyond our focus here) and in general made the background more compact. Nevertheless, we have not cut down the overall content too much, as the thorough background introduction was seen by reviewer TQmq07 as well as by us as a strength of our paper.

---

### Official Review · Reviewer_27e6 · 2024-10-10
**New pruning technique for neural networks**

**Confidence:** 4

**Summary:**

This work proposes a new method for pruning neural networks. This method is based on α gradient-first magnitude-next strategy. The conduct experiments on various models and datasets and prove experimentally that this method beats the state-of-the-art methods. They consider the gradients first, focusing on the parameters with small gradient magnitudes to be omitted, and then they focus on small magnitudes – selecting the parameters with minimal effect on the loss. In more details their method chooses the parameters to prune based on two steps, where first step it ranks the parameters by their gradient magnitudes. Then it selects the smallest of these to prune.

**Strengths:**

* Intuitive method that is well-motivated and presented. The paper is well written and the idea can be conveyed very clearly to the reader.
* The proposed method beats the state-of-the-art pruning methods. It also includes an extensive comparison of the other methods.

**Weaknesses:**

I would say that the ablation study is kind of limited but the paper has limited pages for publication. I would suggest to extend it and include more experiments exploring the proposed method. For example Figure 2 has different sparsity levels but it would be more informative to include lower levels of sparsity too (like 10% and gradually increase to 99%).

**Final Rebuttal Confidence:**

4

**Final Rebuttal Justification:**

I have read all the reviews and comments in the paper and I believe that the paper should get accepted to this venue.

**Justification:**

The paper is good and has a good idea. I strongly believe that the readers will benefit of it.

---

> ### Author Rebuttal · Authors · 2024-10-25
>
> We thank the reviewer for the insightful and encouraging review.
>
> > “Figure 2 has different sparsity levels but it would be more informative to include lower levels of sparsity too (like 10% and gradually increase to 99%).”
>
> Since network compression via pruning typically aims to substantially reduce the number of parameters, often high sparsity ratios are the target, given which we provided results only for sparsity levels >=90%. This is also in line with the experiments presented by other state-of-the-art methods that we compared with. Furthermore, given that the presented results for such sparsity ratios are already comparable to the dense network baseline, lower sparsity ratios would likely yield similar findings, between the dense and highly-sparse results.
>
> > “I would say that the ablation study is kind of limited but the paper has limited pages for publication. I would suggest to extend it and include more experiments exploring the proposed method.”
>
> To provide further insight into the methods, we have now included a new figure (Fig. 2 in the revised version) to demonstrate the distribution of parameter (weight) values of networks trained with a dense model as well as models pruned using our method FGGP and the closest competition GraNet. This new figure helps highlight that both pruning methods help reduce near-zero (ineffective) weights, while our method prunes many more of these.

---

### Meta-Review · Area_Chair_xjsr · 2024-11-01

**Recommendation:** Reject
**Confidence:** 4

**Metareview:**

This work introduces a new criterion for pruning neural networks based on gradient magnitude, asserting that large gradients in weights indicate contribution regardless of the weight magnitude.

The authors propose an innovative two-step approach to address theoretical conjectures on gradient magnitude, aiming to avoid pruning inactive weights too early by focusing only on those that are not actively updating. The paper is well-written, clear, and thorough, with interpretable explanations that support reproducibility.

While the pruning criterion is interesting, there is a consensus among reviewers that the experimental gains are modest and exhibit high variance, which limits the perceived benefits of the proposed method. Although the authors rightly point out that other approaches often yield minimal improvements, this should not be taken as a benchmark. Given the modest empirical improvements demonstrated here and the use of only one dataset, it is challenging to justify the broader applicability of the proposed method. Strengthening the empirical analysis through a more comprehensive study on generalization and pre-convergence pruning performance would improve the rigor and justification of the method.

**Suggested Changes To The Recommendation:**

2: I'm certain of the recommendation.  It should not be changed

---

### Decision · Program_Chairs · 2024-11-06

Reject